# Evaluation of Thermal Fatigue Life and Crack Morphology in Brake Discs of Low-Alloy Steel for High-Speed Trains

**DOI:** 10.3390/ma15196837

**Published:** 2022-10-01

**Authors:** Jinnan Wang, Yunbo Chen, Lingli Zuo, Haiyan Zhao, Ninshu Ma

**Affiliations:** 1State Key Laboratory of Tribology, Department of Mechanical Engineering, Tsinghua University, Beijing 100084, China; 2China Academy of Machinery Science and Technology, Beijing 100044, China; 3Joining and Welding Research Institute, Osaka University, Osaka 567-0047, Japan; 4Beijing National Innovation Institute of Lightweight Ltd., Beijing 100083, China

**Keywords:** brake disc material, crack initiation and propagation, crack growth rate, fatigue failure analysis

## Abstract

Effective braking in high-speed trains is one of the major bottlenecks in expediting the technology and possibilities to improve speed. Although substantial progress has been made to increase operating speed, perhaps, thermal fatigue cracking in brake discs is a primary constraint so far. Thermal fatigue cracking is the major cause of brake disc failure in high-speed trains, especially trains with a speed of 350 km/h or above. In this study, new material composition is proposed for brake discs of high-speed trains. A comprehensive investigation is presented based on fatigue crack initiation and propagation, along with wear and micro-hardness characterization. Thermal fatigue tests at various thermal cycles between 20 ℃ and 700 ℃ were performed and the experimental results are compared with fatigue properties of a commercial brake disc material. An experimental trial revealed that thermal cracks normally initiate and propagate along the oxidized grain boundaries; nevertheless, crack propagation is restricted by the fine precipitates and lath structure of martensitic. Moreover, crack length at the initiation and propagation stage is predicted through crack growth rate and favorable grain size in the crack vicinity. Thermal fatigue life can be improved by dictating the microstructure and precipitate morphology of cast steel by tailoring the alloying composition.

## 1. Introduction

The railway is universally featured as the first rapid land transport means and it is continuously accelerating with recent technological developments in the form of running speed, efficiency, safety, and aesthetics. Over the past century, there has been a significant advancement from conventional locomotives to high-speed bullet trains. Since the brake disc is a safety-critical component of the braking system, the safety evaluation and failure prevention of braking mechanisms for high-speed trains are much more demanding especially when high-speed trains deal with a speed greater than 350 km/h [1]. In a typical braking mechanism, huge kinetic energy is transformed into frictional heat and the subsequent temperature escalates up to 700 ℃ or even higher, which eventually causes plastic strain on the friction surface of the brake disc [2,3]. Essentially, substantial cyclic loading and subsequent localized thermomechanical phenomena inaugurate crack initiation and propagation and gradually leads to failure of the brake disc material in severe braking [4,5,6]. Therefore, it is necessary to study the thermal fatigue properties of potential brake disc material to overcome postulated cracks and corresponding damage mechanisms [7,8]. At present, fatigue cracks are divided into two stages: crack initiation and crack propagation phase. It is difficult to determine the boundary of crack initiation and propagation and to describe the entire evolution process of micro-cracks from initiation to propagation. The thermal fatigue test is an effective method to evaluate the fatigue performance and service life in brake discs of a high-speed train.

Furthermore, a significant advancement in the material design is required for brake discs, as cyclic loading, specifically in the radial direction, inevitably causes thermal cracks due to variations in the physical properties of cyclic loading [9]. The inevitable presence of defects, especially hot shots in the material structure, deteriorates the durability of the material and reduces fatigue life [10]. The prediction of fatigue crack initiation and propagation is very important for the accurate estimation of the material’s fatigue life together with its structural integrity and reliability. The crack initiation and propagation are predicted by a two-phase approach and crack initiation is characterized by a crack size that can be detected by general technical means in a low-cycle fatigue regime [11]. The initial crack sizes of 0.1 and 0.5 mm have been reported previously [12,13]. Under this assumption, a transition crack size is assumed to be amidst the crack initiation and propagation, therefore, linear elastic fracture mechanics can be applied. The crack initiation stage can improve the accuracy of crack life assessment, which was already established by recent experimental studies [14,15]. Baumgartner et al. [16] describe the crack initiation phase with stress or strain-based concepts using an experimental approach in which a 0.5 mm crack size is treated as the crack initiation stage. Heikki Remes et al. [17] studied the crack initiation period by using the strain-based approach. Numerical and experimental results of the fatigue test demonstrated the crack initiation length equal to 0.2–0.5 mm and the critical value of material damage was closely related to the material grain size [17]. An experimental investigation by Murakami et al. [18] revealed the number of stress cycles required to form a grain size crack, is negligibly small in the low-cycle fatigue regime [19]. Zerbst et al. [20] reported the crack is still very short after nucleation, and several grain sizes are selected as the crack initiation length in general research [21,22,23]. Since a newly developed composition is presented to overcome these fatigue performance issues, it is crucial to estimate the thermal fatigue properties and consequences of alloying elements on thermal fatigue performance in potential brake disc materials. The mechanism of crack initiation and propagation at micro and mesoscopic scales was comprehensively analyzed according to the changes in microstructure coarsening, recrystallization, precipitate phase, and crack growth rate during crack propagation.

In the proposed work, the chemical composition of cast steel is tailored to increase fatigue crack resistance during cyclic loading, particularly at elevated temperatures for brake discs of high-speed trains. Thermally-induced crack initiation and propagation mechanisms are examined through an in-situ experimental setup and subsequent cracks are analyzed through SEM (Zeiss Gemini-300, Zeiss, Germany), EDS, Vickers hardness tester (Future-tech FE800, Tokyo, Japan), and XRD (Rigaku D/max-2550, Tokyo, Japan ) analysis for each material composition, such as ZG1, ZG2, and ZG3, individually. The effect of precipitates and their size on crack growth is particularly studied. Moreover, the transition size of the crack from the initiation to propagation is estimated through the crack growth rate and proposed as a tangible correlation with grain size in physical experiments. Thermal fatigue performance is computed by analyzing crack initiation, propagation, crack length, crack tip morphology, and subsequent compositional analysis in a wide range of cyclic loading. Additionally, fatigue and wear properties were studied to apprehend a physical correlation among them.

## 2. Experimental Methods

The ZG2 and ZG3 alloys are the names given to the under-trial newly developed brake disc material. The alloy was smelted with a vacuum induction furnace and subsequently, it was normalized, hardened, and tempered before forming the specimens for fatigue trial. The normalizing temperature was 950 °C, while the quenching and tempering were performed at 920 °C and 600 °C, respectively. The ZG2 and ZG3 specimens were trialed and their performance was compared with commercial brake disc material (ZG1) that has been in use for high-speed train runs at 300 km/h. The compositions of the ZG1, ZG2, and ZG3 materials are listed in Table 1.

According to the design shown in Figure 1a, rectangular plate specimens integrated with a hole on one end and a notch on the other end were fabricated for each material. Fatigue experiments to investigate crack initiation and propagation are executed on thermal fatigue testing apparatus (LRP-1200, JiLin Guanteng Automation Technologyco.,LTD, Jilin, China) by following the Chinese test standard (HB6660-2011). The experimental details are displayed in Figure 1a–c.

The sample was clamped in a fixture and placed outside the resistance furnace using the round hole with 5 mm diameter, as shown in the sample design provided in Figure 1a. The resistance furnace was first heated to 742 ℃ and kept there for 10 min before the sample was placed. The sample was then placed inside the furnace and housed for 55 seconds. When the overall temperature of the sample reached 700 ℃ then it was removed from the furnace and “part B” was immersed in the cooling water for 5 seconds to bring down the temperature, especially at the notched end. An identical process was repeated and samples were tested over the course of 200 cycles for crack length over 4.5 mm and micro-hardness at the notched end. Notched specimens were heated in an air furnace and the notched vicinity was cooled in the water while the remainingpart was cooled down in the air. The subjected water temperature for cooling was maintained at 20 ± 5 °C during the cooling process. A pair of thermocouples were utilized to compute the temperature cycle. Thermocouple A was installed 25 mm away from the notch of the specimen, and thermocouple B was placed near the prefabricated notch, as demonstrated in Figure 1b. Temperature cycle curves at the test positions of the two thermocouples are shown in Figure 2, where t _cooling_ represents the specimen cooling time and t _heating_ is the heating time in thermal cycle. The temperature cycle is further validated with a numerical model established by the finite element method for thermal fatigue life, as demonstrated in Figure 2. The elements adopted in the model were C3D8RT linear hexahedral elements, the element count was 5232, and the number of nodes were 7000 in total. The temperature profile predicted with the FEM simulation is in a good agreement with the results of experiments performed, which shows the reliability of the experimental scheme.

The ZG1, ZG2, and ZG3 alloys were tested with the same experimental trial to evaluate material performance. Each test condition was replicated three times for each material composition to ensure experimental accuracy and reliability. Specimens were carefully observed up to 2000 cycles in each test condition to monitor the temperature and analogous microhardness. Then crack initiation, crack propagation, and crack tip morphology were examined through SEM and EDS analysis. Moreover, XRD analysis was performed on each specimen before and after the trial to obtain a detailed view and subsequent structural variation due to thermal fatigue.

## 3. Results

### 3.1. Thermal Fatigue Crack Propagation and Micromorphology

The relationship between crack length (a) and the number of cycles (N) is obtained by measuring the crack length through an optical microscope. Crack length is observed every 200 cycles and the relevant data is plotted in Figure 3 for under-trial materials. The crack growth rate from small to large was in the order of ZG2, ZG3, and ZG1, possibly due to the crack morphology, microstructure, and mechanical properties. The crack length and crack growth rate of the ZG2 alloy are substantially lower than its competitors in this precise experimental trial.

According to the analysis, the center of the crack was caused by the metal oxidizing during the thermal fatigue phase. Due to the unusual thermal expansion coefficient between metal and metal oxide, the wider the crack, the more metal corrodes via oxidation. As the thermal cycle develops, the metal oxide gradually decreases, and once the old oxide falls off, the exposed metal on the crack propagation surface is oxidized once more.

Figure 4 displays the macroscopic crack morphologies for ZG1, ZG2, and ZG3 after 2000 cycles in the same experimental conditions. The breadth of the crack in ZG2 was comparatively modest and narrow, unlike in ZG1 and ZG3. The analysis revealed that a central crack is formed by the oxidation of the metal during cyclic fatigue; perhaps, the larger the crack width, the more metal is corroded by oxidation owing to the unusual coefficient of thermal expansion in the vicinity. The metal oxide gradually decreases as the thermal cycle progresses, and after the old oxide has fallen off, the exposed metal on the crack propagation surface is oxidized again. 

To some extent, the oxide thickness in cracks reflects the material’s high-temperature oxidation resistance. Based on the narrow crack morphology, ZG2 demonstrated relatively good oxidation resistance as compared to ZG1 and ZG3. The crack tip morphology after the 2000 thermal cycles displayed an obvious contrast for each material composition, as indicated in Figure 4. The crack tip of the ZG1 material is relatively slender and sharp, and there are sharp branches with larger crack propagation and higher oxide layer shedding. Although the tip of the cracks in the ZG2 material has obvious branching, they are all arc-shaped. The oxidation and shedding of material ZG2 on the crack expansion surface are not obvious and are wrapped on the crack expansion surface. This also shows that the oxide of the material is relatively dense, having strong oxidation resistance. The sharpness of the crack tip of material ZG3 is between the other two materials, which is consistent with the a-N curve. The analysis shows that the thermal fatigue resistance of the material was positively related to the high-temperature oxidation resistance of the material. 

### 3.2. Microhardness and Microstructure

The microhardness of the three materials after cyclic loading is shown in Figure 5. The material hardness decreased significantly during the first 200 cycles and then gradually became stable. After 2000 cycles, the hardness of the material decreased from 41.4% to 47.7%, and the properties of the material were significantly affected by thermal cyclic loading. 

The microstructure and analogous precipitates for ZG1, ZG2, and ZG3 are shown in Figure 6, before and after the experimental trial. Before the test, the microstructure composed of high-temperature tempered sorbate and keep the characteristics of martensitic laths, as shown in Figure 6a–c. The martensitic lath structure is more obvious in ZG2 and ZG3 and a good deal of fine and dispersed precipitates are visible in all three materials. The corresponding number and distribution of precipitates are quite identical in ZG1 and ZG2, contrary to ZG3. Resulted microstructure of the three materials is different, this variation is not very obvious before the experiment.

After 2000 thermal cycles, the microstructures of ZG1, ZG2, and ZG3 were transformed due to fatigue, as shown in Figure 6e–g. Apparently, the microstructures show minor martensitic lath formation in ZG1 and ZG2, and perhaps some distinct traits in the case of ZG3.

The size of the precipitates varied significantly, especially in ZG2, where they were uniformly dispersed and rather small and regular in size. The quantity and area of the three precipitates were computed, as shown in Figure 7. The statistical finding showed that the number of precipitates was 145, 391, and 319 for ZG1, ZG2, and ZG3, respectively, within an area between 0 and 1 × 10^4^ nm^2^. It has been well documented that the precipitates in the range of 0–100 nm are called nano-precipitates, and substantially contribute to enhancing material performance [24]. In the process of phase coarsening, the fatigue strength of the alloy decreased gradually, the fatigue crack deflection became more serious, the propagation became more unstable, and the crack growth rate increased.

Variations in precipitate size and distribution dictate subsequent fatigue properties of the material as it substantially depends on precipitate morphology and interaction with plastic deformation bands or micro-cracks in a particular microstructure. Particularly, the ZG1 and ZG3 materials have fairly huge and uneven precipitates in terms of size, however, the nano-precipitates are uniformly tiny and regular in ZG2.

### 3.3. Crack Tip Morphology

The microstructure morphology, element distribution, and precipitates morphology around the crack tip of materials ZG1, ZG2, and ZG3 were compared and analyzed to study their potential cracking resistance. The structure and precipitates of the crack tip were analyzed by SEM and ZG1 and ZG2 demonstrated branching phenomena at the crack tip, as illustrated in Figure 8a,b. The crack tip of ZG1 is sharp, and the angle of the bifurcation is about 45°, while the crack tip and bifurcation are arc-shaped in ZG2. Resistance to cracking with an arc-shaped tip is greater than that of the acute-angle tip with better thermal fatigue resistance. The smaller the angle of the bifurcation, the more likely the metal in the bifurcation would cause stress concentration and accelerate crack propagation. The crack tip area is further enlarged for a more visible view, as shown in Figure 8c,d.

The direction of microcrack propagation at the crack tip of material ZG1 is the same as that of the major crack propagation, according to Figure 8c. The micro-cracks mainly propagate along the austenite grain boundary and the corresponding grains are oxidized, corroded, and even peel off; the propagation path is relatively straightforward after crack growth. Similarly, Figure 8d shows the crack propagation of the material ZG2 is restricted by martensite laths and precipitates. There is a negligible expansion along the grain boundary and a certain angle with the direction of crack propagation is observed. The analysis shows that martensite laths are helpful to reduce both the crack propagation speed and energy. Moreover, Figure 8e shows that the distribution of precipitates is relatively sparse and has little effect on the direction of crack expansion even in the vicinity of the crack. Although the rupture of the precipitate absorbs the crack expansion energy, the subsequent crack tip is still very sharp (Figure 8e). A large number of cracked precipitates are seen along the crack propagation direction. It can be seen that the rupture of the precipitate here not only absorbs the crack propagation energy but also smooths the crack tip, as illustrated in Figure 8f. The precipitates have a certain influence on the crack propagation direction and the subsequent precipitated phase consumes the crack propagation energy on the crack propagation path. 

Thermal fatigue and microstructure affect the material precipitation behavior and the diffraction peaks of the three specimens in the XRD analysis are (110), (200), (211), (022), and (310), as mapped in Figure 9. The combined microstructure of the three materials indicates that the structure is all martensitic and there are no obvious diffraction peaks of the austenite and carbide phases. The XRD results plotted in Figure 9 and Figure 10 are before and after the completion of 2000 cycles in fatigue testing. According to the Scherrer equation [25], the average grain size difference is no more than 10%, which is consistent with the provided results. 

The value of the crystal face with the strongest diffraction peak was (110). The grain orientation for ZG2 was stronger than the other two materials and the directionality of the grain orientation was not conducive to the thermal fatigue resistance of the material [26,27]. The diffraction peaks of the three materials appeared as austenite diffraction peaks and some precipitate peaks, as shown in Figure 10. Thermal fatigue cycles led to the degradation of martensite characteristics, and austenite peaks appeared in ZG1 and ZG3, which can be found in Figure 6 and Figure 10. The subsequent diffraction peaks and the full width at half maxima (FWHM) were calculated. Mainly, the dislocation density of the material was calculated by combining the grain size and micro-strain from the XRD data. The results show that the dislocation density decreased after the thermal fatigue tests by 62.1%, 31.2%, and 5.6% for ZG3, ZG2, and ZG1, respectively. The strongest diffraction intensities for specimens were found to be 93,634, 78,502, and 10,598 accordingly. This shows that the grain orientation directionality of materials increased in the thermal fatigue test. In particular, the grain directionality of ZG3 and ZG1 increased, which compromised the thermal fatigue resistance of the material. The total content of alloying elements in the material was less than 5% and the carbon content was 0.2%; therefore, the diffraction peaks of carbide precipitates are not obvious.

Furthermore, an EDS analysis was performed to observe and diagnose the composition of precipitates in a second phase (Figure 11). It can be seen that there are four main forms of precipitates shown here, which are small circular precipitates, small rod precipitates, massive precipitates, and large strip precipitates. The composition test of precipitation from cultivation form is shown in Table 2.

It can be noticed that the main components of the precipitates are C, Cr, Mn, Mo, and V. The main precipitated phases are MX, MC, M23C6, and M7C3, where M is Cr, Fe, Mn, Mo, and V, and X is C, N, and the vacancy in the proposed material for brake discs of a high-speed train.

### 3.4. Fracture Behavior

Essentially, the brake disc material must have anti-oxidation and sufficient corrosion performance for various environmental conditions, such as high temperature, rain, ice, and snow. Studying the effect of alloying elements on the passivation of thermal fatigue crack growth tips can provide theoretical support for the anti-fatigue design of materials. The energy dispersive spectrometer analysis of the crack tip composition was carried out for ZG2, as shown in Figure 12. Precise element distribution around the crack tip was observed to explore the cause of passivation.

As previously shown in Figure 9, there was a bifurcation phenomenon that manifests as divarication in the front region of the cracks of the two materials. The bifurcation of materials has certain crack propagation, and the crack tip of material ZG1 was relatively sharper as compared to ZG2. The crack region had the appearance of an arc shape in ZG2. The rolling area is a passivation phenomenon of the crack, which helped to reduce crack propagation. The analysis showed that a passivation stabilization phenomenon existed at the crack tip of material ZG2, which is supportive to improve the thermal fatigue crack resistance and decelerate crack growth rate. The EDS analysis showed that the content of chromium is relatively higher in ZG2 and the oxidation resistance was better than in ZG1 and ZG3. Moreover, oxidation corrosion and oxide peeling of the crack surface could be prevented after the thermal fatigue crack was initiated.

### 3.5. Wear Property

Brake discs undergo friction, wear, and thermal fatigue phenomenon simultaneously, and the braking mechanism transforms the train’s kinetic energy into friction and subsequent wear mass loss. The results showed a considerable wear mass loss for the corresponding brake disc materials, as provided in Table 3. It has been established that fine dispersion precipitates can improve the wear properties along with fatigue resistance. Conclusively, well-structured fine precipitates have a positive correlation with friction, wear, and fatigue resistance.

## 4. Discussion

Service conditions affect the internal structure of brake disc material, particularly at high temperatures, which gradually lowers the fatigue resistance [28]. Furthermore, with the dislocations in the slats and on the slat boundaries, martensitic phases emerged and rearranged to reduce the dislocation density. As the dislocation density decreased, subsequent strength and microhardness also declined; likewise, the hardness decreased as the carbon content of the martensitic phase reduced due to nano precipitates. Hardness deteriorated as the ferrites were developed due to the disappearance of twins particularly in the acicular martensitic phase, however they retained their acicular morphology. Chromium content improved the anti-oxidation and anti-corrosion ability of the crack boundaries and passivated the crack tip. It also promoted fatigue crack resistance of a material and restricted crack growth rate. Excessive precipitation should be avoided in material preparation and in later heat treatment phase, specifically, the precipitation of the Fe_3_C phase should be avoided. High chromium content can form a dense film on the cracked surface to abandon further deterioration of the material surface. Since thermal fatigue crack propagation is mainly intergranular in the brake disc material, high chromium can improve the crystallization strength and thermal fatigue resistance. 

The lath structure of the martensite is helpful to improve the fatigue crack resistance of the material. The increase of carbide types may reduce the growth of carbides, and perhaps the proper adjustment of the alloying element ratio can control over precipitation of carbides as well as the formation of metal compounds. 

The thermal fatigue crack growth model presented in this study is illustrated in Figure 13. The precipitate phase presumably absorbs crack propagation energy and curtails crack growth rate. Subsequently, this would slow down crack initiation and propagation in the presence of a favorable grain structure. Previous studies have established that the smaller the size of the precipitates, the better the effect of the dispersion material. Moreover, the greater the crack and dissolution resistance of the material precipitates, the stronger the fatigue resistance of the material [29,30]. Nonetheless, the ZG2 precipitated phase is fine, regular, and uniform, perhaps stronger than that of ZG1, which is consistent with its softening resistance. The analysis believes that the larger the number of small-size precipitates, the better the strengthening effect of the material, and the better the fatigue resistance and softening resistance. The smaller the size difference of the precipitated phase of the material, the better it is to improve the uniformity of the material performance and the thermal fatigue resistance of the material. The more uniform distribution of the precipitate phase encourages material stability and thermal fatigue resistance. The better the anti-dissolution and anti-aggregation growth ability of the precipitated phase, the better it helps to resist deterioration of material performance.

From the microscopic point of view, the most common explanation of metal crack formation is slip band cracking. With the increase of loading cycles, the corresponding dislocation density of the crystals also increases. When the dislocation density reaches a threshold value, dislocation entanglement is supposed to form inside the crystal and results in high and low-density dislocation bands, which hinders the dislocation movement in the regime of interest. Under the continuous fatigue loading, dislocations interact with each other and shifted towards a low-energy direction. This gradually accumulatesdislocation cells, which then develop into the sub-crystal structure. In this way, the evolution and inner movement of dislocations within the crystal cause slip bands within the metal. The occurrence of slip lines, the development of slip bands, and the formation of resident slip bands motivate to outset slip bands. Dislocation movement first arises on the weak grain inside the metals and eventually results in traces on the metal surface, called slip lines. These slip lines aggregate and gradually form slip bands in continuous cyclic loading. When the slip band is continuously squeezed into and out of the grain boundary by thermal cycles, the slip band is transformed into a resident slip band. Traces are left by resident slip bands on the surface of the material, and when such traces are applied deep enough, the initial crack is formed. Therefore, the resident slip band is the key factor in crack formation.

The microstructure plays an important role in the fatigue performance of the brake disc material [31]. The fatigue strength of an alloy gradually decreases in the process of phase coarsening and recrystallization. As far as crack deflection occurs, it leads to it crack propagation and ultimately stimulates a higher crack growth rate.

## 5. Conclusions

The proposed work is focused on crack initiation and propagation, fatigue failure, and wear characteristics of a newly developed brake disc material for high-speed trains. Disc material (ZG2) is developed by tailoring the material composition of cast steel and its properties are further compared with commercially available ZG1 over 2000 cycles. The experiments revealed that brake disc failure mechanisms take place at elevated temperatures during a higher number of loading cycles. The following conclusions are drawn from a comprehensive experimental trial.
(1)Introducing additional elements in cast steel adequately hinders crack growth rate and yields favorable precipitates to overcome fatigue cracking. Chromium and vanadium contents can develop a dense film structure on the crack propagation surface to prevent further deterioration of the fracture surface;(2)Materials with better anti-dissolution and anti-aggregation ability in the precipitates have more resistance against cracking and fatigue failure. The destruction of precipitated phases consumes crack propagation energy and subsequently passivates the crack tip;(3)The hardness of steel material for brake discs changes in the same way. It shows that it quickly softens in the early stage of the cycle and stabilizes after the cycle;(4)The microstructure significantly dictates fatigue properties and crack formation ultimately affects material performance. Fine, homogenous, uniform, and regular microstructures provide resistance to alleviate crack growth rate;(5)The precipitate size, uniform distribution, anti-dissolution, and anti-aggregation growth ability contribute to fatigue resistance.


Research outlook on crack initiation and the propagation mechanism based on multi-scale probability statistics and machine science may establish a unified model and standard. This study provides theoretical support and data accumulation for improving the thermal fatigue resistance in brake disc materials for high-speed bullet trains.

## Figures and Tables

**Figure 1 materials-15-06837-f001:**
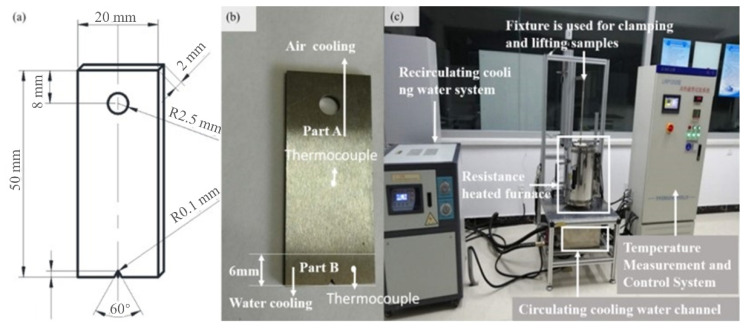
Schematic diagram of dimensions and experimental facilities: (**a**) schematic diagram of dimensions; (**b**) sample pictures; (**c**) thermal fatigue testing apparatus.

**Figure 2 materials-15-06837-f002:**
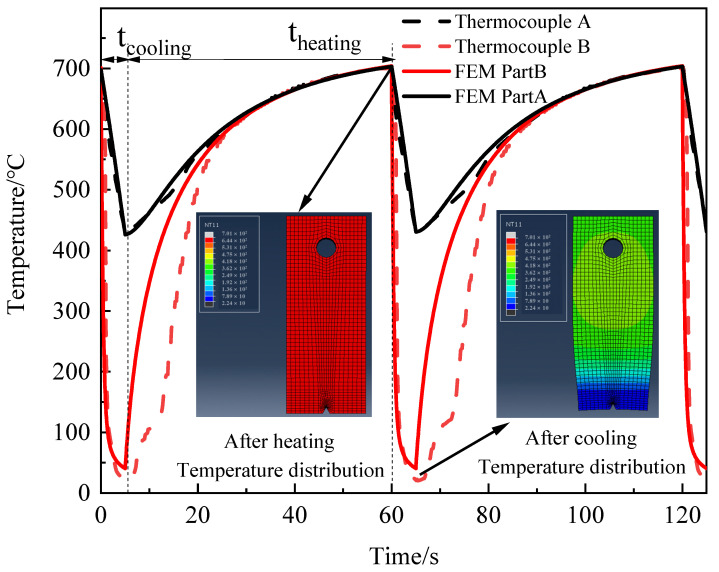
The time-temperature curve for cyclic loading at the notch and 25 mm away from the notch.

**Figure 3 materials-15-06837-f003:**
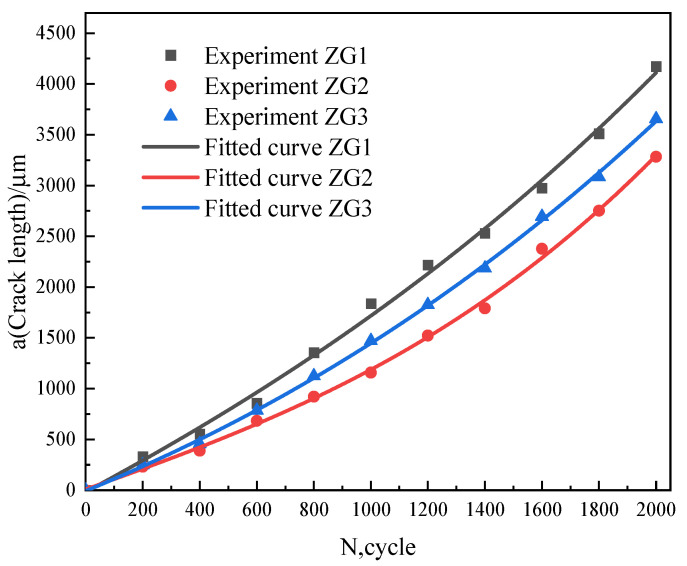
The a-N curves of the thermal fatigue cycle.

**Figure 4 materials-15-06837-f004:**
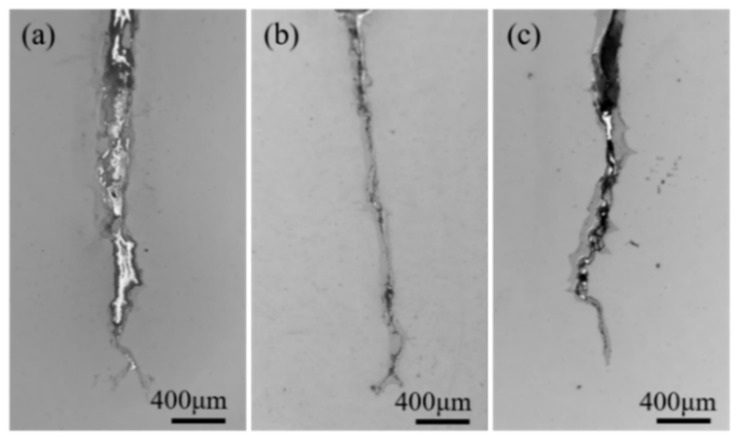
Crack morphology of three materials after 2000 cycles: (**a**) ZG1; (**b**) ZG2; (**c**) ZG3.

**Figure 5 materials-15-06837-f005:**
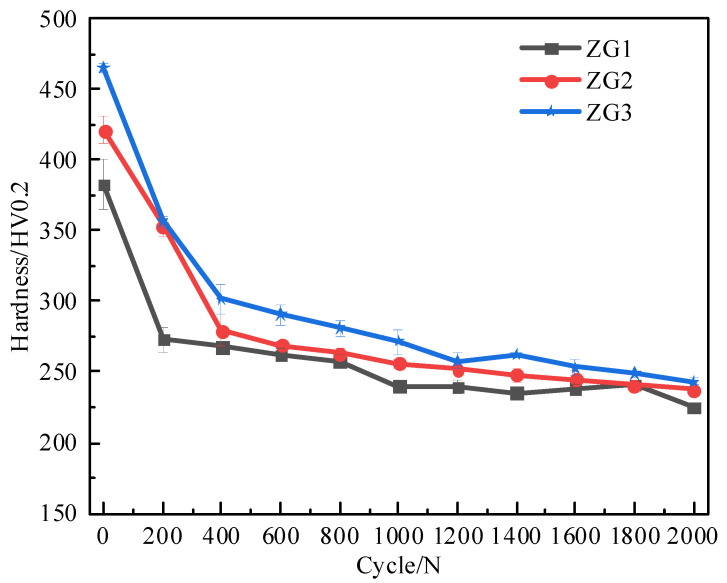
Material hardness variation with the number of thermal cycles.

**Figure 6 materials-15-06837-f006:**
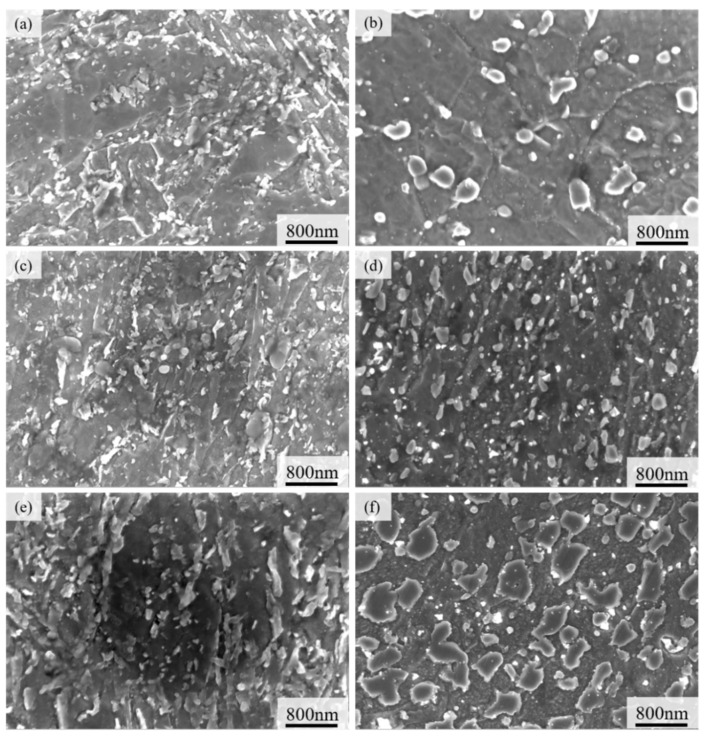
Precipitation phase and precipitates morphology before and after thermal fatigue cycle: (**a**) ZG1-pre; (**b**) ZG2-pre; (**c**) ZG3-pre; (**d**) ZG1-after; (**e**) ZG2-after; (**f**) ZG3-after.

**Figure 7 materials-15-06837-f007:**
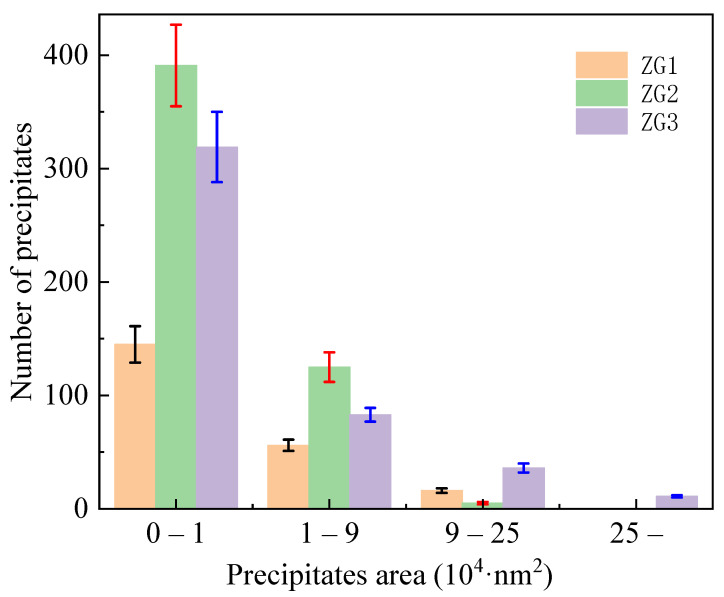
Statistics of the number and area size of precipitates after the thermal fatigue cycle.

**Figure 8 materials-15-06837-f008:**
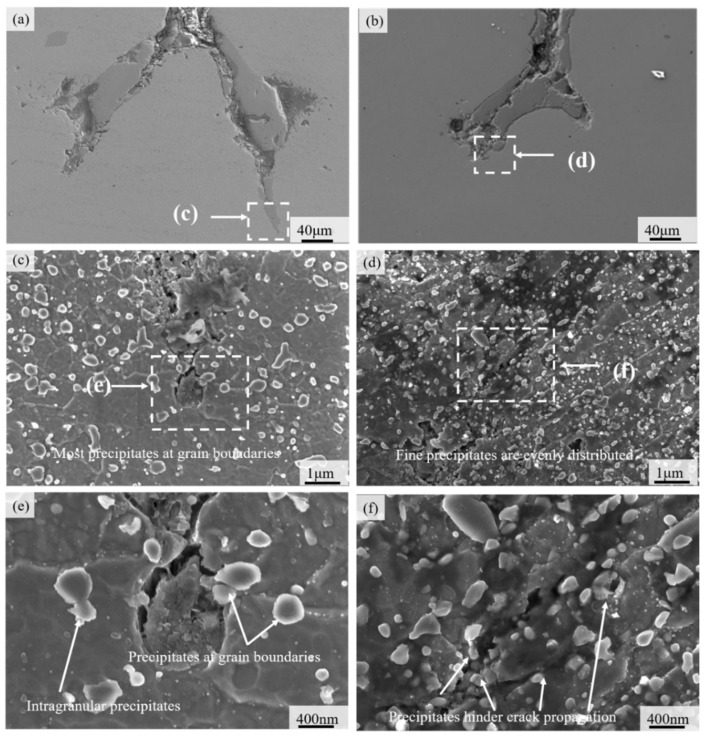
The topography of crack tip: (**a**) crack tip morphology of material ZG1; (**b**) crack tip morphology of material ZG2; (**c**) crack tip of material ZG1; (**d**) crack tip of material ZG2; (**e**) precipitated phase near the crack tip of ZG1; (**f**) precipitated phase near the crack tip of ZG2.

**Figure 9 materials-15-06837-f009:**
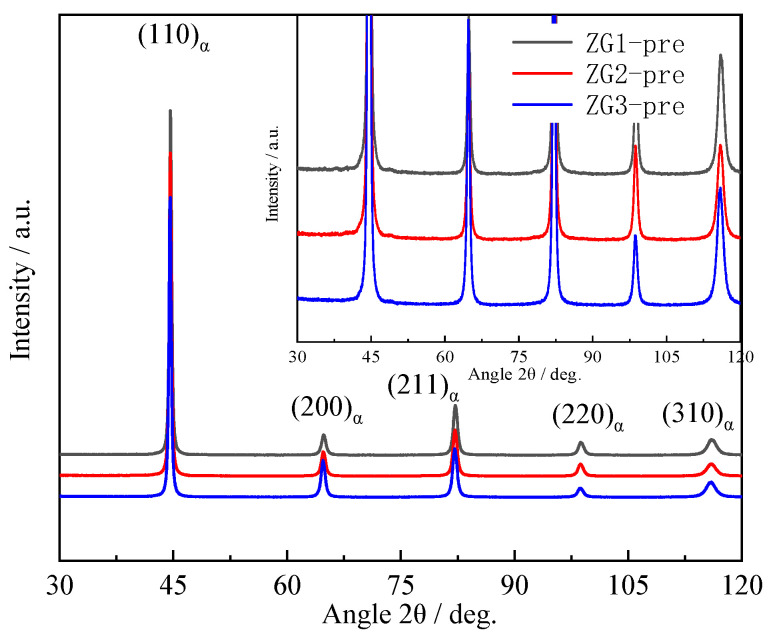
X-ray diffraction profiles of specimens before a thermal fatigue test.

**Figure 10 materials-15-06837-f010:**
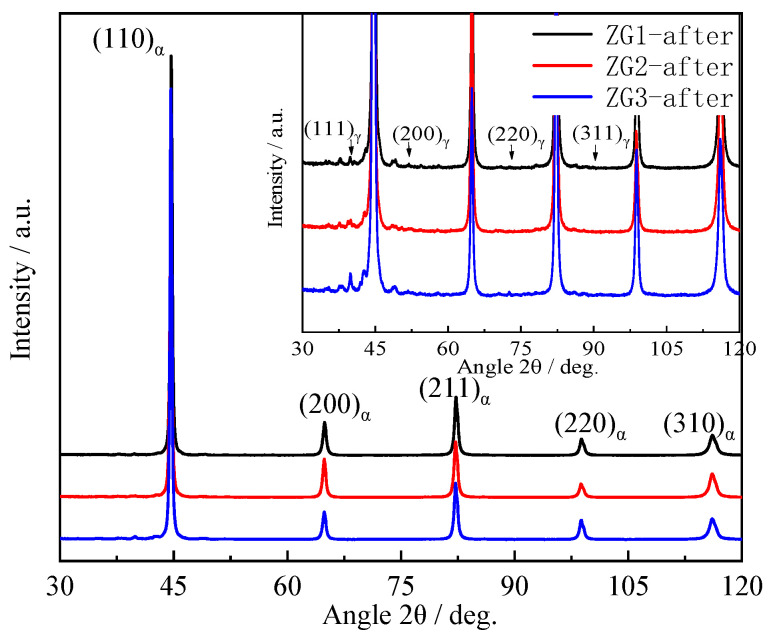
X-ray diffraction profiles of specimens after thermal fatigue test.

**Figure 11 materials-15-06837-f011:**
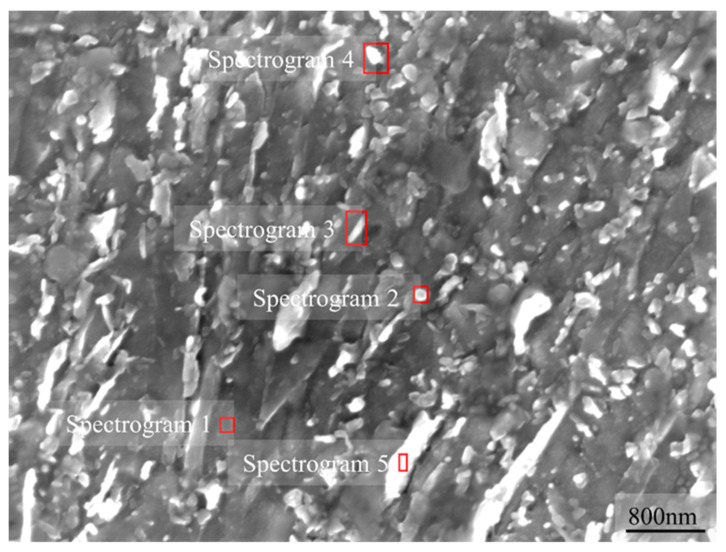
EDS spectrum test electron chart.

**Figure 12 materials-15-06837-f012:**
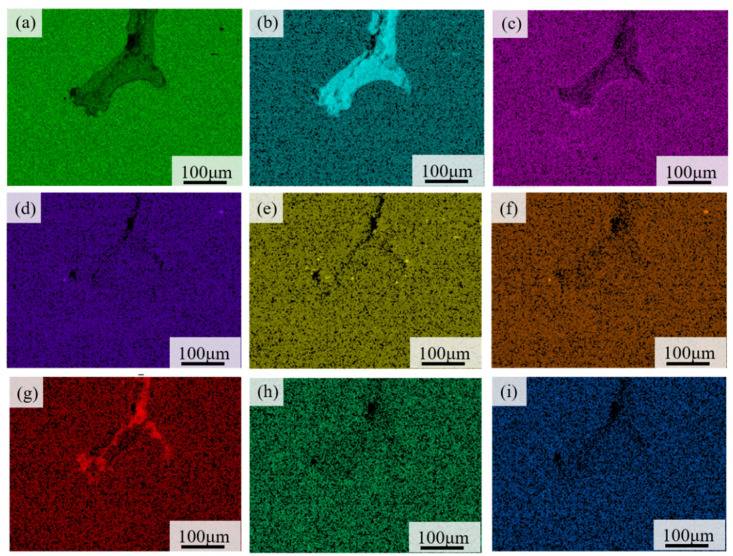
The element distribution map at the crack tip of material ZG2: (**a**) Fe; (**b**) O; (**c**) Cr; (**d**) Mo; (**e**) Si; (**f**) Mn; (**g**) C; (**h**) V; (**i**) Ni.

**Figure 13 materials-15-06837-f013:**
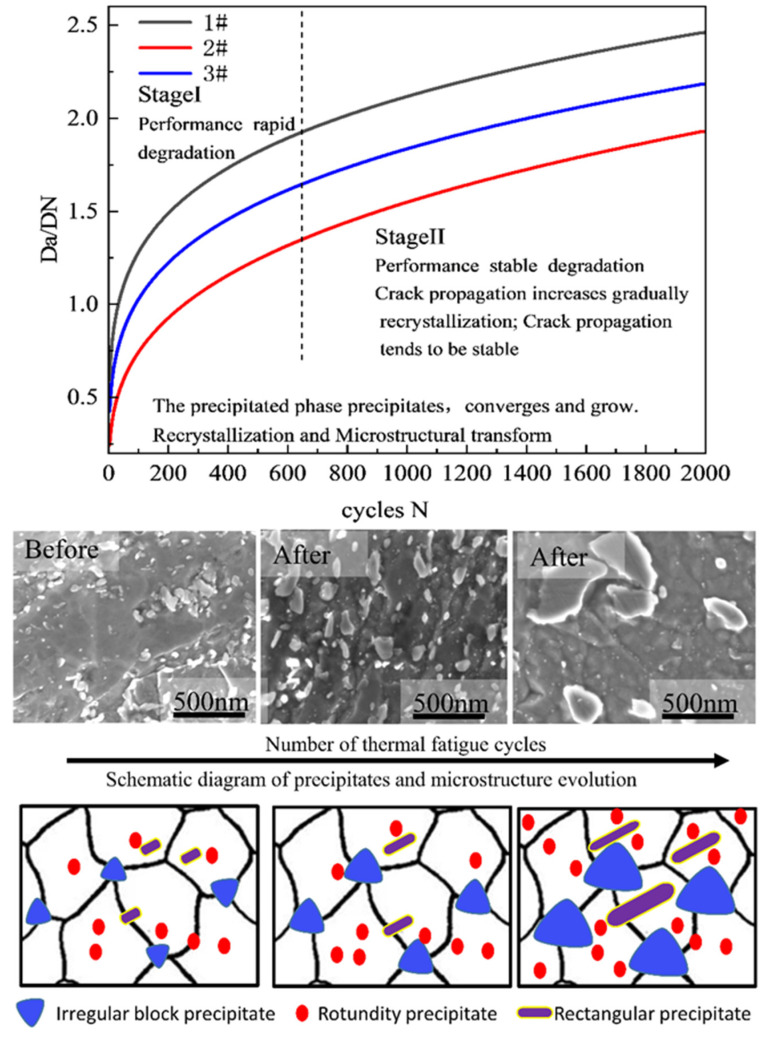
Thermal fatigue cracks propagation model and schematic diagram of precipitate and microstructure evolution.

**Table 1 materials-15-06837-t001:** Chemical compositions (wt%) of the cast steels ZG1, ZG2, and ZG3.

NO	C	Si	Cr	Mo	Ni	Mn	V	Fe
ZG1	0.22	0.41	0.80	1.0	0.95	1.0	-	Bal.
ZG2	0.30	0.50	1.60	0.80	0.95	1.0	0.1	Bal.
ZG3	0.30	0.50	1.60	1.20	0.95	1.0	-	Bal

**Table 2 materials-15-06837-t002:** Composition of precipitates (at %).

NO\Element	Fe	C	Mn	Cr	Ni	Si	Mo	V	O
Point 1	71.77	18.37	0.65	6.84	0.3	0.67	0.44	0.11	0.84
Point 2	70.13	13.80	0	8.97	0.45	0.63	0.27	0.1	5.66
Point 3	70.64	19.00	1.85	6.64	0.60	0.71	0.39	0.17	0.9
Point 4	71.48	17.89	2.64	4.72	0.57	0.66	0.34	0.12	1.59
Point 5	72.04	15.56	3.79	5.23	0.6	0.63	0.37	0.21	1.46

**Table 3 materials-15-06837-t003:** Wear mass loss (mg) of design and service materials at different rotational speeds.

Linear Speed/(m/s)	0.4	0.8	1.2	1.6	2.0	Total
ZG1	19.4	28.6	52.7	46.9	15.7	163.3
ZG2	26.5	16.1	34.5	53.3	19.3	149.7
ZG3	27.6	32.6	57.8	62.3	27.2	207.5

## Data Availability

Not applicable.

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
