# Peer review of "Evaluation of Thermal Fatigue Life and Crack Morphology in Brake Discs of Low-Alloy Steel for High-Speed Trains"

_materials, 2022, doi:10.3390/ma15196837_

Round 1
Reviewer 1 Report
General
· Language has to be improved; it is strongly recommended to re-look the ‘Grammar’ and ‘Presentation’ thoroughly by an expert before re-submission.
Points in favor
o The present article may be considered as add-on information in the relevant domain.
Points detracting
How the number of precipitates was counted? How much frame was considered? What was the error bar? The number of particles should be in unit area/ volume – the Y-axis unit has to be re-looked!
What was matrix grain size and how they have contributed in deformation study?
What was the nature of precipitates – I think there are more than one type of second phases as Cr and Mo were present in substantial quantity with adequate carbon. Have they equally contributed in the evolution of microstructure during experimentation - Otherwise who played the dominant role? Presentation on evolution of microstructure in sequence under stress and temperature schematically may help in conceiving the essence of the fact.
When the volume fraction or size distribution of second phase become small, they cannot be identified by conventional XRD or may be presented with feeble intensity (Fig.9 and 10) where a lot of imagination is involved (hence chance of error becomes many fold), which authors also indicated; in such situation to confirm their presence a different technique may be adopted to strengthen the inference.
Was there any stress-induced precipitation for the investigated systems?
Somewhere (L.247) a statement has been given on relative change in dislocation density for different specimens; such type of statements should be supported by experimental data (especially, when X-ray diffraction has been carried out).
The brake disc material experiences severe wear and tear; in that respect authors are suggested to furnish the data on same and compare with existing materials to bring novelty in manuscript.
How the authors in present situation compare the actual state of repetitive thermal exposure and stress in cyclic manner during service exploitation with experimental parameters for brake disc material – a justification should be furnished to bring convergence in ‘scope of investigation’!
1. Decision
The manuscript in its present form is not upto the standard of the journal; hence needs ‘major revision’.
Reviewer 2 Report
This paper studies the problem of modeling the thermal fatigue life and crack morphology in the low alloy steel for high-speed trains. At first glance, the manuscript is interesting. This paper explores the implementation study with a new and real perspective on the current scientific problem. The article is well written and the topic could be of interest to the readers of the journal. The study is important in its applied sense. The research presented in this article seeks novelty. The choice of research methodology is appropriate and logical. Keywords and sources from the cited literature are suitable.
Some comments are listed below.
- What are the new findings of the present manuscript? What research gap did you find from previous researchers in your field? The actuality and novelty of the current problem must be clarified (it is still partially described, but needs to be expanded and made clearer). Mention it in the Introduction section. It will improve the strength of the article.
- Figure 2 shows a comparison of FEM calculations with experimental data. The FE model is not explained. The FE model should be described in more depth and a detailed interpretation of the results should be provided.
- The conclusion section of the paper is very condensed. If the authors' conclusions provided a more interesting contribution to the motivation for this study and some interesting open questions for future research, it would encourage the reader to take more interest in the topic.
Overall, my opinion is positive and I would suggest that the authors continue the work.
